# Genetic Uniqueness and Genetic Structure of Populations of *Picea obovata* Ledeb. and *Larix sibirica* Ledeb. in the Northern and Middle Urals

Andrei Zhulanov [1], Nikita Chertov [1], Yulia Nechaeva [1], Viktoriia Pechenkina [1], Larisa Zhulanova [1], Svetlana Boronnikova [1,*] and Ruslan Kalendar [2,3,*]

1 Faculty of Biology, Perm State University, Perm 614990, Russia; aumakua.ru@gmail.com (A.Z.); nikita.chertov22@gmail.com (N.C.); ulia-2012@mail.ru (Y.N.); p_viktoria2@mail.ru (V.P.); larisa_zhulanova@mail.ru (L.Z.)
2 National Laboratory Astana, Nazarbayev University, Nur-Sultan 010000, Kazakhstan
3 Helsinki Institute of Life Science HiLIFE, University of Helsinki, Biocenter 3, Viikinkaari 1, 00014 Helsinki, Finland
* Correspondence: svboronnikova@yandex.ru (S.B.); ruslan.kalendar@helsinki.fi (R.K.); Tel.: +358-294-158-869 (R.K.)

**Abstract:** Establishing sustainable plantations with genetic diversity equivalent to that of natural populations is vital for successful reforestation efforts. In this study, we present an innovative approach for selecting populations suitable for reforestation, taking into account their genetic uniqueness using Inter Simple Sequence Repeats (ISSR) markers. Our investigation focused on six populations of *Picea obovata* Ledeb and six populations of *Larix sibirica* Ledeb, collected from the Northern and Middle Urals. We found that the indicators of genetic diversity were significantly higher in *L. sibirica* compared to *P. obovata*, while the number of rare alleles was greater in Siberian spruce (R = 19). Among the *P. obovata* populations, the Cherdyn's forestry exhibited notably high genetic diversity, and for *L. sibirica*, the Gainy's forestry stood out in this regard. Moreover, the genetic subdivision of the six *P. obovata* populations ($G_{ST}$ = 0.331) was higher than that of the six *L. sibirica* populations ($G_{ST}$ = 0.177). To ensure optimal seed selection considering the genetic originality coefficient (GOC) and population differentiation, we recommend utilizing the *P. obovata* population from Gainy's forestry with a GOC of 0.554 and the *L. sibirica* population from Cherdyn's forestry with a GOC of 0.372. These populations harbor typical alleles characteristic of the research region, making them ideal candidates for seed selection. Furthermore, the specific alleles identified can serve as valuable markers for determining the geographic origin of *P. obovata* and *L. sibirica* wood, aiding in efforts to trace the sources of these species in forestry and trade practices.

**Keywords:** genetic differentiation; genetic originality; genetic structure; inter simple sequence repeats (ISSR); *Larix*; *Picea*; typical alleles; Ural





## 1. Introduction

Most forest woody plants have a significant level of intraspecific genetic variability, which is found mainly within populations [1,2]. A decrease in intraspecific variability reduces the adaptive potential of plants [3,4]. Anthropogenic influences' conversion of forest into other land uses has a negative impact on woody plants due to genetic degradation, reduction, and fragmentation of their habitats, leading to a decrease in the total and effective abundance and density of populations, up to the disappearance of individual local populations. Deforestation leads to the loss of unique genotypes, which inevitably leads to genetic impoverishment of populations and their degradation [5].

At present, the main problem of preserving the unique genetic material of woody plants for the selection of populations for reforestation has not been solved.

In the Perm Territory, which is located in the Northern and Middle Urals, the total area of the forest fund is 12,005.0 thousand hectares, which is 71.3% of the territory of the region. The predominant coniferous species in the Perm Krai is spruce, which occupies 61% (95,234.5 thousand ha) of the area of all coniferous woody plants in the region, and larch occupies only 0.2% (14,500 ha) [6].

The Perm Krai is one of the leading logging regions of Russia. The spruce wood is massively cut down, since Perm enterprises produce about 20% of the all-Russian volume of paper for various purposes. The dense wood of larch (density 660–700 kg/m$^3$) is especially appreciated, as it is also resistant to moisture. In this regard, larch wood is in great demand and is also cut down, sometimes illegally; thus, it is necessary to study the genetic diversity of the existing populations of this species in order to develop measures for its conservation.

The control of the geographic origin of wood is directly related to the study of genetic diversity, showing the differentiation of populations, as well as a detailed genetic description of plantations, forests, or areas where wood could be harvested, which can be used in judicial investigations related to the illegal use of forests [7].

Siberian spruce (*Picea obovata* Ledeb., family *Pinaceae*) is one of the dominant species of dark coniferous forests in Russia. Its range extends from the north of the European part of Russia to the Pacific coast. It is stated that Siberian spruce in its pure form grows in the North of the European part of Russia, in the Northern Urals, and in Siberia up to the Sea of Okhotsk coast [6]. In the northern and northwestern parts of the Perm Krai, spruce has an almost continuous range.

Most researchers today agree that on the territory of the East European Plain, a large complex of spruce populations of hybrid origin has formed as a result of the processes of introgressive hybridization of European spruce (*P. abies*) and Siberian spruce (*P. obovata*) [8].

Several authors regard the natural hybrids between these two spruce species as a distinct species, namely *P. fennica* (Regel) Kom [9]. According to the theory of introgressive hybridization put forward by E.G. Bobrov, transitional forms between *P. abies* and *P. obovata*, common in Eastern Europe and the Urals, are the result of a long hybridogenic interaction between European and Siberian species.

The distribution areal of *P. fennica* covers most of the Russian Plain between the eastern boundary of *P. abies* and the western part of *P. obovata*, as well as most of Scandinavia (Finland, Sweden, and Norway), except for its extreme north, also including the North, Middle, and South Urals [10,11]. It is noted that it is difficult to draw the exact boundary of their distribution, as many morphological characters flatten to the periphery of the range; the term "spruce complex *Picea abies—P. obovata*" is currently used [12–15].

Species of the genus *Larix* Mill. are common woody plants in Russia and the planet as a whole. Larch forests are of great economic, ecological, and biospheric importance [16]. Species of the genus *Larix* Mill. belong to light coniferous plants [17,18].

Within the Urals region, the genus *Larix* is represented by the western species of the Siberian larch *Larix sibirica* Ledeb. (family *Pinaceae*). Extensive research conducted over several years across the Urals, including the Perm Krai, has revealed a noticeable fragmentation of larch plantations. Furthermore, a distinct geographical feature known as the "leafless wedge" characterizes the western macro-slope of the Ural Mountains within this area [19–21].

Currently, various types of molecular genetic markers are used to study the genetic diversity and spatial differentiation of tree forms of individual plants [22].

Molecular genetic markers play a crucial role in diverse applications, including genome mapping, disease diagnosis, and classifying individuals or populations [23]. These markers can be specific genes or sequences with unknown functions. Currently, the analysis of genetic polymorphisms in genome sequences is typically performed using several approaches: detection of polymorphisms using various PCR-based genome profiling techniques; the use of hybridization systems on various platforms and microarrays; and the use of next-generation sequencing (NGS) or Third Generation Sequencing (TGS) for comprehensive analysis [24,25]. Molecular genetic markers are becoming widely used in



environmental forensics. Determining the origin of timber is particularly important for identifying illegally logged and marketed timber. Intrapopulation genetic diversity and population differentiation of Norway spruce and Siberian spruce were described in detail earlier using isoenzyme analysis [26–28], including in the Southern Urals.

The molecular genetic diversity of conifers (*P. abies, P. fennica, P. obovata* (*Pinaceae*)) has previously been studied using multiple-loci polymorphic DNA markers based on PCR-based genome profiling applications, for example, using random amplification of polymorphic DNA (RAPD) [29] and Inter Simple Sequence Repeats (ISSR) [30]. Efficient use of single-loci polymorphic DNA markers based on PCR-based methods for the detection of Simple Sequence Repeats (SSR) to study populations of *Picea abies* (L.) H. Karst. and *Picea obovata* Ledeb has been carried out [27,31]. Thus, data on the variability of nuclear microsatellite SSR loci differentiate well the southern and northern groups of European populations of *Picea abies* [32].

In the Perm Krai, the genetic diversity of three spruce species (*Picea abies* (L.) Karst., *Picea obovata* Ledeb. and *Picea fennica* (Regel) Kom) was studied fragmentarily using only ISSR PCR-based multiple-loci polymorphic DNA markers for genome profiling applications [30]. The ISSR PCR analysis was used to assess the genetic diversity and genetic structure of the populations [33]. This method allows for a large number of polymorphic DNA fragments that are reproduced during repeated PCR to be received. Neutral molecular markers as an ISSR application for PCR-based genome profiling are successfully used in population genetics, and have emerged as valuable tools for assessing genetic variation and differentiation in plant populations [34].

The genetic variability of natural larch populations in the Urals was studied by V.P. Putenikhin and Z. Kh. Shigapov et al. [21,35,36] using isoenzyme markers, as well as in the Subpolar Urals and on the eastern macro-slope of the Ural Mountains by V. L. Semerikov et al. [37] using isoenzyme, mitochondrial, chloroplast, AFLP markers, and detection of nucleotide polymorphisms of some potentially adaptive-significant genes. The highest level of species diversity in the Urals was noted in the "Permian-Kama Cis-Uralian" population, which was established using a complex of data from morphological and isoenzyme analyzes [38]. Highly informative PCR-based genome profiling applications, such as SSR, AFLP, RAPD, and ISSR methods, are now effectively used to study the genetic polymorphism of conifers and, in particular, species of the genus *Larix* [39–43]. Wide-scale surveys of the evolutionary history and distribution of the genus *Larix* using mitochondrial and chloroplast genetic markers have been conducted on 116 populations in Russia, Mongolia, Kazakhstan, and China [37].

Coniferous woody plants contain various Natural Bioactive Compounds (NBC), such as terpenoids, steroids, alkaloids, flavonoids, and others, as well as resin acids, which are extensively utilized in the pulp and paper industry and are prospective natural bioactive compounds [44–46].

Currently, there is a discussion on issues of forest seed zoning (see "On the establishment of forest seed zoning" dated 8 October 2015 (as amended on 28 March 2016)) and approaches to selecting populations to preserve the unique gene pool. The classification of the diversity of genotypes based on the results of molecular marking with the determination of GOC (genetic originality coefficient) was proposed by E. K. Potokina and T. G. Aleksandrova for herbaceous plant species [47]. This approach was applied by us in relation to the study of woody plant species [48]. Determination of GOC in the widespread species *P. obovata* for the selection of populations for reforestation in areas of intensive timber harvesting in the Northern and Middle Urals has not been carried out before. The aim of this study was to evaluate genetic diversity and originality of the genetic pool and genetic structure of populations of coniferous plant species (species of the genera *Picea* and *Larix*) for selection of populations for reforestation in the Northern and Middle Urals.

## 2. Materials and Methods

Six natural populations of Siberian spruce (*Picea obovata* Ledeb., Pinaceae) growing in the north and northeast of the Perm Krai in places of timber harvesting were selected as objects of study. The studied samples of *P. obovata* in Perm Krai were located in the areas of the following forestries: *Po_Ch*—Cherdyn's forestry; *Po_Gn*—Gainy's forestry; *Po_Kr*—Krasnovishersk's forestry; *Po_Br*—Berezniki's forestry; *Po_Kg*—Sivin's forestry; and *Po_Os*—Kungur's forestry (Table 1 and Figure 1). The *Po_Gn*, *Po_Kr* and *Po_Ch* populations are located in the Northern Urals, while the *Po_Br*, *Po_Kg* and *Po_Os* populations are located in the Middle Urals. The largest geographic distance (355 km) was noted between the studied populations of *Po_Kr* and *Po_Os P. obovata*, and the smallest (72 km) between the populations of *Po_Kr* and *Po_Ch* (Table S1).

**Table 1.** The studied natural populations of *P. obovata* and *L. sibirica* used in the ISSR analysis.

| Population [a] | Location | Number of Trees | Coordinates |
|---|---|---|---|
| | *Picea obovata* L. | | |
| *Po_Gn* | Perm Krai, Gainy's forestry | 31 | N: 60.3411 E: 53.8403 |
| *Po_Ch* | Perm Krai, Cherdyn's forestry | 31 | N: 60.4335 E: 56.3076 |
| *Po_Kr* | Perm Krai, Krasnovishersk's forestry | 30 | N: 60.3839 E: 57.6145 |
| *Po_Br* | Perm Krai, Berezniki's forestry | 31 | N: 59.3463 E: 57.0734 |
| *Po_Kg* | Perm Krai, Sivin's forestry | 31 | N: 58.3546 E: 54.7026 |
| *Po_Os* | Perm Krai, Kungur's forestry | 31 | N: 57.3117 E: 55.9024 |
| | *Larix sibirica* Ledeb. | | |
| *Ls_Ih* | Perm Krai, Vishersky Reserve | 30 | N: 61.1178 E: 59.1537 |
| *Ls_Tl* | Perm Krai, Vishersky Reserve | 30 | N: 61.1130 E: 58.8899 |
| *Ls_Bn* | Perm Krai, Cherdyn's forestry | 30 | N: 60.5147 E: 55.9203 |
| *Ls_Kr* | Perm Krai, Krasnovishersk's forestry | 30 | N: 60.3264 E: 57.0931 |
| *Ls_Gn* | Perm Krai, Gainy's forestry | 30 | N: 60.1739 E: 53.6213 |
| *Ls_Kh* | Sverdlovsk Region, Karpinsk's forestry | 30 | N: 58.7824 E: 59.4068 |

[a] International designations of geographic latitude: N—north latitude; E—east longitude.

The comparison group included six samples from the populations of the western race of Siberian larch (*Larix sibirica* Ledeb., family *Pinaceae*) located in the same territory of the Perm Krai and one population in the neighboring Sverdlovsk Region. The studied samples of *L. sibirica* were located in: *Ls_Tl* and *Ls_Ih*—Vishersky Reserve, *Ls_Bn*—Cherdyn's forestry, *Ls_Kr*—Krasnovishersk's forestry, *Ls_Gn*—Gainy's forestry, and *Ls_Kh*—Karpinsk's forestry of the Sverdlovsk region (Table 1 and Figure 1). The *Ls_Tl*, *Ls_Ih*, *Ls_Bn*, *Ls_Kr*, and *Ls_Gn* populations are located in the Northern Urals, and the *Ls_Kh* population is located in the Middle Urals.

The greatest geographical distance (361 km) was noted between the studied *Ls_Gn* and *Ls_Kh L. sibirica* populations, and the shortest (14 km) between the *Ls_Tl* and *Ls_Ih* populations located on different mountain peaks (Table S2).

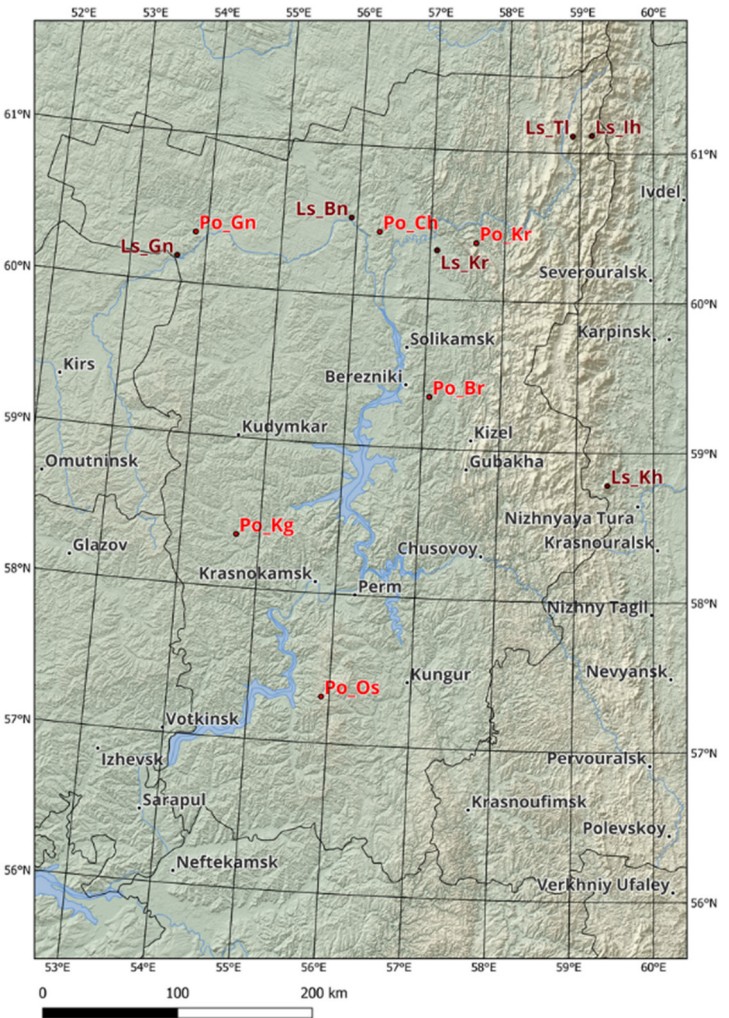

**Figure 1.** Schematic map of the location of the studied populations of *P. obovate (Red color)*: *Po_Gn*—Gainy's forestry, *Po_Cn*—Cherdyn's forestry, *Po_Os*—Kungur's forestry, *Po_Kr*—Krasnovishersk's forestry, *Po_Br*—Berezniki's forestry, *Po_Kg*—Sivin's forestry; designation of *L. sibirica* populations (Brown color): *Ls_Tl* and *Ls_Ih*—Vishersky Reserve, *Ls_Bn*—Cherdyn's forestry, *Ls_Kr*—Krasnovishersk's forestry, *Ls_Gn*—Gainy's forestry; in the Sverdlovsk Region, *Ls_Kh*—Karpinsk's forestry.

Three species of the genus *Picea* are distributed in the Perm Krai: *P. abies*, *P. obovata*, and *P. fennica* [49]. According to the forest-seed zoning, *P. obovata* dominates in the study region. When collecting material, the species affiliation of trees was determined by morphological features (Table S3).

*L. sibirica* is characterized by solitary microstrobili and paired ovules consisting of spirally arranged microsporophylls, symmetrically located with holes (micropyle) down the upper side of the scales [19].

Needles were taken from each of the 185 trees from 6 populations of *P. obovata* and from the 180 trees of 6 populations of *L. sibirica* for DNA isolation. In each population, the material was collected from 30–31 trees located at a distance of at least 100 m from each other.

DNA isolation was carried out according to a modified method using acid CTAB (Cetyl Trimethyl Ammonium Bromide) [50]. To determine the concentration and quality of DNA we used a Spectrofotometer™ NanoDrop 2000 ("Thermo Scientific", Waltham, MA, USA).

A 25 μL reaction mixture was used for PCR reactions. Each individual reaction mixture consisted of 25 ng of template DNA, 1× PCR buffer supplemented with 2.5 mM $MgCl_2$, 0.2 μM ISSR primer, 0.2 mM of each dNTP, and 1 U of Taq DNA polymerase obtained from Sintol (Moscow, Russia).

The PCR amplification was performed in a SimpliAmp™ thermocycler (Thermo Fisher Scientific Inc., Waltham, MA, USA) according to the conditions as follows: initial denaturation at 94 °C for 2 min, followed by 32 cycles of amplification at 94 °C for 20 s, temperature increased from 55 to 65 °C (depending on the primer sequence) [51] for 30 s, and then 72 °C for 60 s. A final elongation step at 72 °C for 3 min (Table 2) was performed.

**Table 2.** Characteristics of ISSR primers used to assess genetic diversity of *P. obovate* and *L. sibirica*.

| ID | Sequence 5′-3′ | Tm (°C) * | GC (%) | Ta (°C) |
|---|---|---|---|---|
| CR-212 $(CT)_8TG$ | CTCTCTCTCTCTCTCTTG | 52.6 | 50.0 | 56 |
| CR-215 $(CA)_8GT$ | CACACACACACACACAGT | 58.1 | 50.0 | 56 |
| ISSR8 $(GAG)_6C$ | GAGGAGGAGGAGGAGGAGC | 64.3 | 68.4 | 56 |
| M1 $(AC)_8CG$ | ACACACACACACACACCG | 61.0 | 55.6 | 56 |
| M3 $(AC)_8CT$ | ACACACACACACACACCT | 59.5 | 50.0 | 56 |
| X9 $(ACC)_6G$ | ACCACCACCACCACCACCG | 68.6 | 68.4 | 64 |
| X10 $(AGC)_6C$ | AGCAGCAGCAGCAGCAGCC | 70.0 | 68.4 | 64 |
| X11 $(AGC)_6G$ | AGCAGCAGCAGCAGCAGCG | 70.1 | 68.4 | 64 |

* Tm (°C): melting temperature; Ta (°C): optimal annealing temperature; GC (%): percentage of guanine-cytosine content.

The ISSR primers were developed by Kalendar et al. [52], and previously selected effective ISSR primers for *Pinus sylvestris* and *P. sibirica* (Table 2) were used [33]. Effective ISSR primers have also been previously identified for *P. obovata* [30] and for *L. sibirica* [53]. For two species (*P. obovata* and *L. sibirica)*, two ISSR primers were common, while one of them had two nucleotides in the core motif (CR-215 with the formula $(CA)_6GT$), and the other had three nucleotides (primer "X10" with the formula $(AGC)_6C$). In addition to the common two primers, three primers for *P. obovata* (M1 with formula $(AC)_8CG$; CR-212 $(CT)_8TG$; "X9" $(ACC)_6G$, Table 3) and three primers for *L. sibirica* (M3 with formula $(AC)_8CT$; "X11" $(AGC)_6G$; ISSR8 $(GAG)_6C$) were used.

**Table 3.** Genetic diversity of six *P. obovata* populations.

| Populations | *He* | $n_a$ | $n_e$ | *I* | *R* |
|---|---|---|---|---|---|
| *Po_Kr* | 0.146 (0.016) | 1.553 (0.499) | 1.225 (0.282) | 0.232 (0.023) | 2 |
| *Po_Ch* | 0.178 (0.018) | 1.675 (0.470) | 1.295 (0.351) | 0.276 (0.025) | 4 |
| *Po_Gn* | 0.144 (0.016) | 1.605 (0.491) | 1.227 (0.310) | 0.230 (0.023) | 2 |
| *Po_Br* | 0.162 (0.017) | 1.605 (0.491) | 1.264 (0.333) | 0.251 (0.025) | 6 |
| *Po_Kg* | 0.138 (0.017) | 1.500 (0.502) | 1.226 (0.326) | 0.214 (0.024) | 2 |
| *Po_Os* | 0.108 (0.015) | 1.456 (0.500) | 1.169 (0.275) | 0.174 (0.022) | 3 |
| Total | 0.146 (0.007) | 1.983 (0.132) | 1.344 (0.318) | 0.230 (0.010) | 19 |

*He*—expected heterozygosity; $n_a$—absolute number of alleles per locus; $n_e$—effective number of alleles per locus; *I*—Shannon index; for all the above parameters, standard deviations are given in brackets; *R*—number of rare fragments.

The genetic diversity of *P. obovata* and *L. sibirica* was evaluated using PCR amplification for DNA profiling with all effective ISSR primers. Following amplification, PCR products were electrophoresed in a 1.5% agarose gel with 1xTBE buffer at 70 V for 5 h. After separation, the gel was stained with ethidium bromide and imaged under ultraviolet UVA light using the GelDoc XR gel documentation system from BioRad (Hercules, CA, USA). To determine the length of DNA fragments, a molecular weight marker (100 bp + 1.5 + 3 KbDNALadder, LLC SibEnzim-M, Moscow, Russia) and the QuantityOne program (Bio-Rad Laboratories, Inc., Hercules, CA, USA) were used. In total, polymorphism of ISSR markers with 5 primers was analyzed in 185 probes of DNA of *P. obovate*,

and in 180 probes of DNA of *L. sibirica*. For two species (*P. obovata* and *L. sibirica*), two ISSR primers were common, and three primers were separate for each species (Figures S1–S17).

The data underwent computer processing using the dedicated VBA-macro GenAlEx designed for MS Excel [54]. This processing aimed to ascertain several parameters, including the count of alleles ($n_a$), effective ($n_e$) number of alleles, proportion of polymorphic loci ($P_{95}$), expected (*He*) heterozygosity, and Shannon's information index (*I*).

The genetic structure of populations was described using the following parameters calculated in the POP-GENE 1.31 program [54]: the expected proportion of heterozygous genotypes in the whole population as a measure of total genetic diversity ($H_T$); the expected proportion of heterozygous genotypes in a subpopulation as a measure of intrapopulation diversity ($H_S$); the proportion of interpopulation genetic diversity in total diversity or the gene differentiation coefficient ($G_{ST}$) [55].

The coefficient of genetic originality was determined according to the method proposed for herbaceous plants [47], modified by us for woody plants [48].

To establish the correlation between pairwise genetic distances ($D_N$) and geographic distances within the general population group, we employed the widely recognized Mantel test. For each sample, we obtained 19 basic climatic parameters from the WorldClim service's bioclimatic variables database, utilizing the raster v3.4–13 package [56,57]. This information was used to create a distance matrix through Canberra distance calculations. To examine the correlation between genetic and climatic distances, we conducted a Mantel test using the GenAlEx MS Excel tool. Additionally, we employed Principal Coordinates Analysis (PCA) via the PAST 4.10 program to validate the collected data. In the PAST 4.10 program [58], we generated an intricate dendrogram for all trees using the Neighborjoining method, and we further analyzed and visualized the data using the UMAP (Uniform Manifold Approximation and Projection) method [59].

### 3. Results

#### 3.1. Genetic Diversity and Genetic Uniqueness of P. obovata and L. sibirica Populations

Molecular genetic analysis of 6 populations of *P. obovata* in the Northern and Middle Urals (Perm Krai) revealed 115 DNA fragments (Table S4), of which 89 were polymorphic, so the proportion of polymorphic loci ($P_{95}$) in the total sample was 0.774. In 6 populations of *L. sibirica*, 114 DNA fragments were found, of which 101 were polymorphic (Table S5). In this regard, the proportion of polymorphic loci in the total sample (180 trees) of *L. sibirica* was higher ($P_{95} = 0.886$) than in the total sample (185 trees) of *P. obovata* ($P_{95} = 0.774$). Using the Mann—Whitney test, it was shown that the differences in this indicator between the species were significant (*p*-value < 0.050).

The length of DNA fragments amplified by PCR in *P. obovata* varied from 200 to 3000 bp. In this species, the largest proportion of polymorphic loci (0.862) was identified using primer CR-212 with the formula $(CT)_8TG$, and the smallest (0.600) was determined using primer "X10" $(AGC)_6C$ (Table S4). The largest proportion of polymorphic loci in *P. obovata* ($P_{95} = 0.763$) was observed in the *Po_Gn* population, and the smallest ($P_{95} = 0.710$) in the *Po_Os* population. But the differences in this indicator between the populations were not significant (*p*-value > 0.050). In *L. sibirica*, the length of DNA fragments amplified by PCR varied in a smaller range, from 200 to 1370 bp. In this species, the largest proportion of polymorphic loci (0.913) was identified using primer "X10" $(AGC)_6C$, and the smallest (0.833) was determined using primer CR-215 with the formula $(CA)_8GT$ (Table S5).

The proportion of polymorphic loci in *L. sibirica* was higher ($P_{95} = 0.868$) in the *Ls_Bn* population, while the lowest ($P_{95} = 0.759$) was in the *Ls_Kh* population. But the differences in this indicator between the populations were not significant (*p*-value > 0.050). An analysis of the genetic diversity indicators (*He*, *I*, $n_a$, $n_e$) in *P. obovata* showed that all of them have a maximum value in the *Po_Ch* population, and a minimum value in the *Po_Os* population (Table 3). Differences in *He*, *I*, and $n_e$ between these populations were significant (*p*-value < 0.050). The *Po_Ch* population is located in typical growth conditions for the

species, while the *Po_Os* population is remote and isolated from other populations of the species.

When analyzing the same indicators of genetic diversity ($He$, $I$, $n_a$, $n_e$) in *L. sibirica*, the maximum values were noted in the *Ls_Gn* population, and the minimum values were noted in the *Ls_Tl* population (Table 4), which is located in the mountains at an altitude of 680–800 m. Differences in terms of $He$, $I$, and $n_e$ between populations were unreliable ($p$-value > 0.050).

**Table 4.** Genetic diversity of six *L. sibirica* populations.

| Populations | $He$ | $n_a$ | $n_e$ | $I$ | $R$ |
|---|---|---|---|---|---|
| Ls_Ih | 0.202 (0.018) | 1.640 (0.482) | 1.341 (0.368) | 0.307 (0.026) | 0 |
| Ls_Tl | 0.194 (0.019) | 1.558 (0.494) | 1.331 (0.375) | 0.292 (0.026) | 0 |
| Ls_Bn | 0.225 (0.018) | 1.649 (0.479) | 1.381 (0.363) | 0.338 (0.026) | 0 |
| Ls_Kr | 0.210 (0.017) | 1.649 (0.479) | 1.345 (0.341) | 0.321 (0.025) | 1 |
| Ls_Gn | 0.243 (0.018) | 1.693 (0.463) | 1.414 (0.369) | 0.364 (0.026) | 1 |
| Ls_Kh | 0.233 (0.015) | 1.690 (0.460) | 1.391 (0.361) | 0.352 (0.025) | 4 |
| Total | 0.218 (0.007) | 1.921 (0.271) | 1.426 (0.311) | 0.329 (0.011) | 6 |

$He$—expected heterozygosity; $n_a$—absolute number of alleles per locus; $n_e$—effective number of alleles per locus; $I$—Shannon index; for all the above parameters, standard deviations are given in brackets; $R$—number of rare fragments.

When comparing the indicators of genetic diversity of the two species, it was found that they were higher for the total sample of *L. sibirica* ($P_{95}$ = 0.886; $He$ = 0.218; $n_e$ = 1.426; $I$ = 0.329), compared with the total sample of *P. obovata* ($P_{95}$ = 0.774; $He$ = 0.146; $n_e$ = 1.344; $I$ = 0.230). Using the Mann—Whitney test, it was found that the indicators of $He$, $I$, and $n_e$ for the total sample of *L. sibirica* differed significantly from the same indicators for the total sample of *P. obovate* ($p$-value < 0.050).

To determine genetic diversity at the population level, rare DNA fragments (occurring with a frequency of less than 5%) are important. In the total sample of *P. obovata*, 19 rare DNA fragments were noted. At the same time, their maximum number ($R$ = 6) was noted in *Po_Br*, and in other populations—from 2 to 4 rare DNA fragments. In the total sample of *L. sibirica*, only 6 rare alleles were noted. The maximum number of rare alleles was noted in the population *Ls_Kh* (R = 4), in *Ls_Kr* and *Ls_Gn*—1 allele each. Three populations of *L. sibirica* lacked rare alleles. To select populations for seed collection for further reforestation, it is not enough to select only populations with high genetic diversity. It is necessary to select populations with typical and specific alleles for the study region.

The approach to determining the genetic originality coefficient (GOC), proposed specifically for markers with dominant inheritance [47], was modified [48] for woody plants, using species of the genus *Populus* as an example.

Among the 6 populations of *P. obovata* studied (Table S6), the minimum GOC values were noted in the *Po_Gn* (GOC = 0.554) and *Po_Ch* (GOC = 0.676) populations, i.e., these 2 populations are carriers of typical alleles for the study region and have typical gene pools. In addition, the *Po_Ch* population had the highest genetic diversity. High GOCs were observed in *Po_Kr* (GOC = 0.886) as well as *Po_Os* (GOC = 1.026). These populations are carriers of specific alleles for the study region and have specific gene pools.

In *L. sibirica*, the lowest GOC values were noted (Table S7) in the *Ls_Bn* (GOC = 0.372) and *Ls_Kr* (GOC = 0.432) populations. These 2 populations have a typical gene pool and have typical alleles. The *Ls_Kh* and *Ls_Gn* populations had high GOCs of 0.783 and 0.661, respectively. These populations have a specific gene pool and have specific alleles. The *Ls_Gn* population has the highest levels of genetic diversity.

We identified two populations with high and low GOC values for each species. In order to select a smaller number of populations, that is, one with a typical and one with a specific gene pool, it is necessary to study and select populations for reforestation to take into account the differentiation of populations in the study area.

Thus, genetic diversity is higher for the total sample of *L. sibirica* ($P_{95}$ = 0.886; *He* = 0.218; $n_e$ = 1.426; *I* = 0.329) when compared to the total sample of *P. obovata* ($P_{95}$ = 0.774; *He* = 0.146; $n_e$ = 1.344; *I* = 0.230), apart from the number of rare DNA fragments in *P. obovata* (R = 19). In *P. obovata*, the maximum indicators of genetic diversity were noted in the *Po_Ch* population, and the minimum in the *Po_Os* population. In *L. sibirica*, the maximum values were noted in the *Ls_Gn* population, and the minimum values in the *Ls_Tl* population.

According to the GOC determination, two populations with low GOC values were identified for each species. To select a smaller number of populations, it is necessary to study and select populations for reforestation, taking into account the differentiation of populations in the study area.

*3.2. Genetic Structure and Differentiation of Six Populations of P. obovata and Six Populations of L. sibirica*

Analysis of the genetic structure of the studied 6 populations of *P. obovata* showed (Table 5) that the highest index of population subdivision ($G_{ST}$ = 0.413) was detected using primer X9 (ACC)$_6$G, and the lowest ($G_{ST}$ = 0.260) was detected using primer M1 (AC) $_8$CG. For the total sample of *P. obovata*, the index of subdivision of populations was 0.331.

**Table 5.** Genetic structure and differentiations of six populations of *P. obovata*.

| Primer ID | $H_T$ | $H_S$ | $G_{ST}$ |
|---|---|---|---|
| M1 (AC)8CG | 0.224 (0.027) | 0.166 (0.016) | 0.260 |
| CR-212 (CT)8TG | 0.212 (0.027) | 0.132 (0.007) | 0.375 |
| CR-215 (CA)6GT | 0.233 (0.025) | 0.165 (0.015) | 0.291 |
| X10 (AGC)6C | 0.181 (0.031) | 0.112 (0.011) | 0.381 |
| X9 (ACC)$_6$G | 0.232 (0.030) | 0.136 (0.014) | 0.413 |
| Total | 0.218 (0.027) | 0.146 (0.013) | 0.331 |

$H_T$—expected proportion of heterozygous genotypes in the entire population; $H_S$—expected proportion of heterozygous genotypes in a subpopulation; $G_{ST}$—the share of interpopulation genetic diversity in the total diversity or an indicator of subdivision of populations; standard deviations are given in brackets.

For *P. obovata* samples, no significant correlation ($R^2$ = 0.0529, *p* = 0.16) between genetic and geographic distances was found (Figure S18), but there was a significant positive correlation ($R^2$ = 0.5576, *p* = 0.03) between genetic and climatic distances (Figure S19). Matrices of pairwise $F_{ST}$ values and a dendrogram of *P. obovata* genetic similarity were constructed and visualized (Table 6, Figure 2).

**Table 6.** Pairwise genetic distances between the studied populations of *P. obovata*.

| Po_Kr | Po_Ch | Po_Gn | Po_Br | Po_Kg | Po_Os | |
|---|---|---|---|---|---|---|
| 0.000 | 0.380 | 0.465 | 0.363 | 0.504 | 0.484 | Po_Kr |
| 0.163 | 0.000 | 0.293 | 0.306 | 0.334 | 0.454 | Po_Ch |
| 0.205 | 0.100 | 0.000 | 0.378 | 0.212 | 0.369 | Po_Gn |
| 0.155 | 0.124 | 0.151 | 0.000 | 0.425 | 0.434 | Po_Br |
| 0.217 | 0.109 | 0.052 | 0.167 | 0.000 | 0.443 | Po_Kg |
| 0.203 | 0.187 | 0.112 | 0.177 | 0.137 | 0.000 | Po_Os |

The largest genetic distances (Table 6) were noted between the populations of *P. obovata* *Po_Kr* and *Po_Gn* ($D_N$ = 0.217), and the smallest between *Po_Gn* and *Po_Kg* ($D_N$ = 0.052).

The clustering of genetic distances between *P. obovata* trees was performed using the UMAP method (Figure 3). Four clusters were distinguished—three were formed by the populations *Po_Kr*, *Po_Br*, and *Po_Os*, and the fourth one combines trees from the populations *Po_Ch*, *Po_Gn*, and *Po_Kg*. Analysis of the genetic structure of six studied populations of *L. sibirica* showed (Table 7) that the highest index of population subdivision ($G_{ST}$ = 0.320) was detected using primer CR-215 (CA)$_6$GT, and the lowest ($G_{ST}$ = 0.101) was detected using primer X11 (AGC)$_6$G. For the total sample of *L. sibirica*, the index of population differentiation was 0.177.

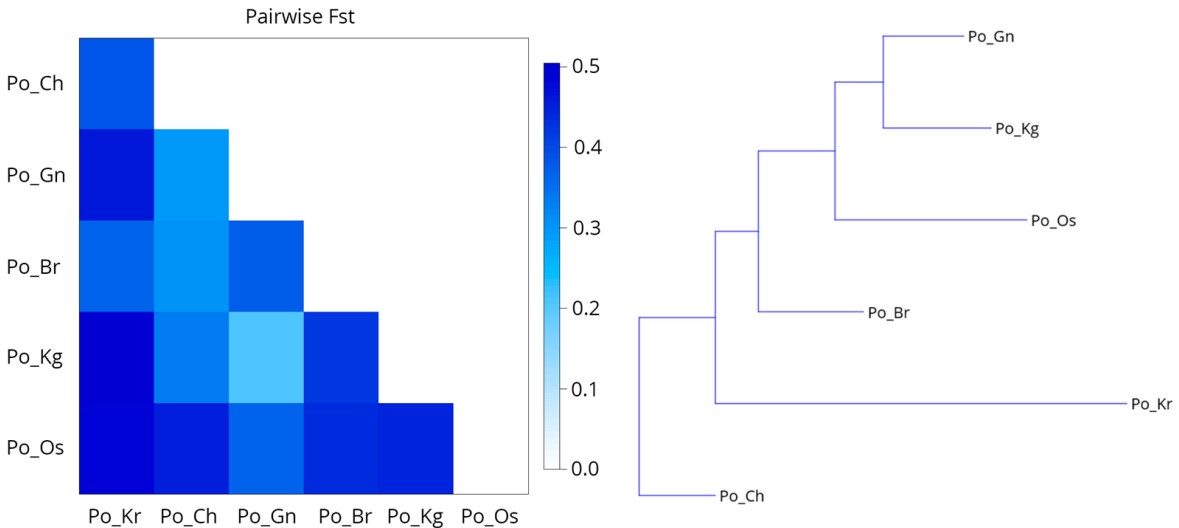

**Figure 2.** Visualization of pairwise $F_{ST}$ values between *P. obovata* populations and a dendrogram based on these values.

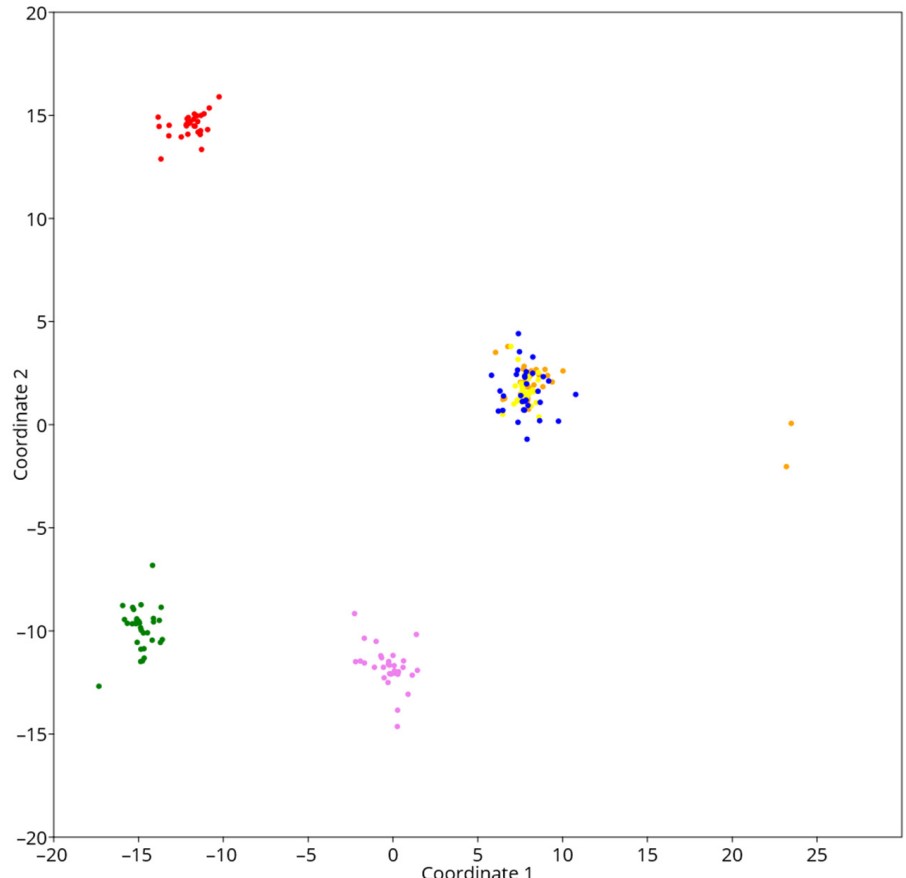

**Figure 3.** Clustering of the studied *P. obovata* trees using the UMAP method. Red dots—population *Po_Kr*, orange dots—population *Po_Ch*, yellow dots—population *Po_Gn*, green dots—population *Po_Br*, blue dots—population *Po_Kg*, purple dots—population *Po_Os*.

**Table 7.** Genetic structure and differentiations of six populations of *L. sibirica*.

| Primers | $H_T$ | $H_S$ | $G_{ST}$ |
|---|---|---|---|
| M3 (AC)8CT | 0.260 (0.026) | 0.220 (0.021) | 0.155 |
| X11 (AGC)6G | 0.278 (0.019) | 0.250 (0.016) | 0.101 |
| CR-215 (CA)6GT | 0.219 (0.035) | 0.149 (0.018) | 0.320 |
| X10 (AGC)6C | 0.282 (0.028) | 0.235 (0.024) | 0.167 |
| ISSR8 (GAG)6C | 0.268 (0.026) | 0.197 (0.016) | 0.262 |
| Total | 0.264 (0.025) | 0.218 (0.019) | 0.177 |

$H_T$—expected proportion of heterozygous genotypes in the entire population; $H_S$—expected proportion of heterozygous genotypes in a subpopulation; $G_{ST}$—the share of interpopulation genetic diversity in the total diversity or an indicator of subdivision of populations; standard deviations are given in brackets.

For *L. sibirica* populations, there was a significant positive correlation ($R^2$ = 0.8685, $p < 0.05$) between genetic and climatic distances (Figure S20). A mean positive correlation ($R^2$ = 0.3574, $p < 0.05$) was also found between the genetic and geographical distances of the studied populations of *L. sibirica* (Figure S21).

Matrices of pairwise $F_{ST}$ values and a dendrogram of the genetic similarity of *L. sibirica* were built and visualized (Table 8, Figure 4). The largest genetic distance (Table 8) was observed between the populations *Ls_Tl* and *Ls_Gn* ($D_N$ = 0.149), and the smallest between *Po_Bn* and *Po_Kr* ($D_N$ = 0.028). The clustering of genetic distances between *L. sibirica* trees was performed using the UMAP method (Figure 5).

**Table 8.** Pairwise genetic distances between the studied populations of *L. sibirica*.

| *Ls_Tl* | *Ls_Ih* | *Ls_Bn* | *Ls_Kr* | *Ls_Gn* | *Ls_Kh* | |
|---|---|---|---|---|---|---|
| 0.000 | 0.079 | 0.285 | 0.258 | 0.276 | 0.151 | *Ls_Tl* |
| 0.035 | 0.000 | 0.261 | 0.240 | 0.258 | 0.158 | *Ls_Ih* |
| 0.148 | 0.139 | 0.000 | 0.041 | 0.116 | 0.257 | *Ls_Bn* |
| 0.127 | 0.122 | 0.028 | 0.000 | 0.062 | 0.220 | *Ls_Kr* |
| 0.149 | 0.143 | 0.065 | 0.038 | 0.000 | 0.224 | *Ls_Gn* |
| 0.071 | 0.079 | 0.154 | 0.124 | 0.136 | 0.000 | *Ls_Kh* |

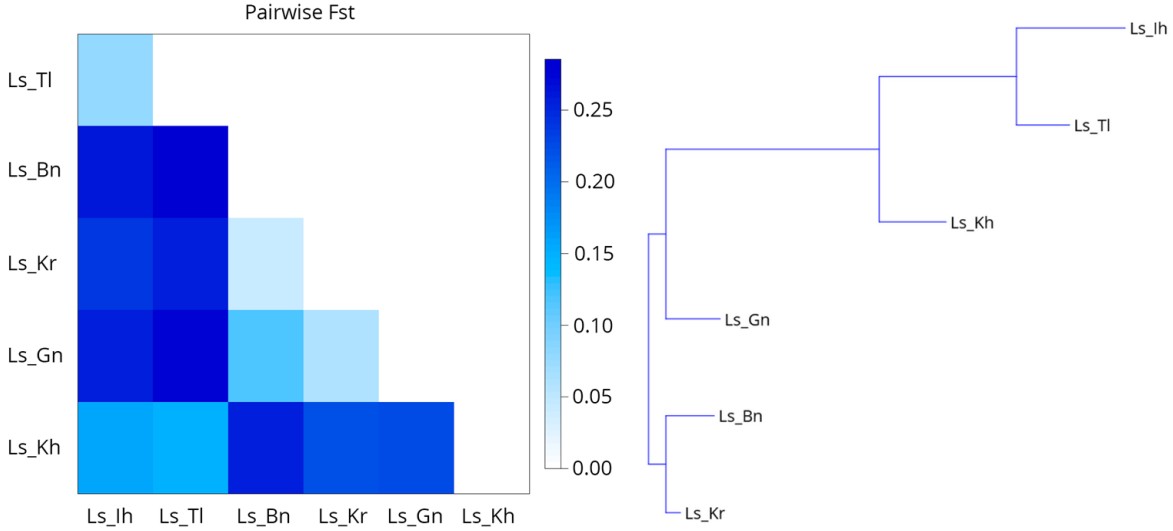

**Figure 4.** Visualization of pairwise $F_{ST}$ values between *L. sibirica* populations and a dendrogram based on these values.

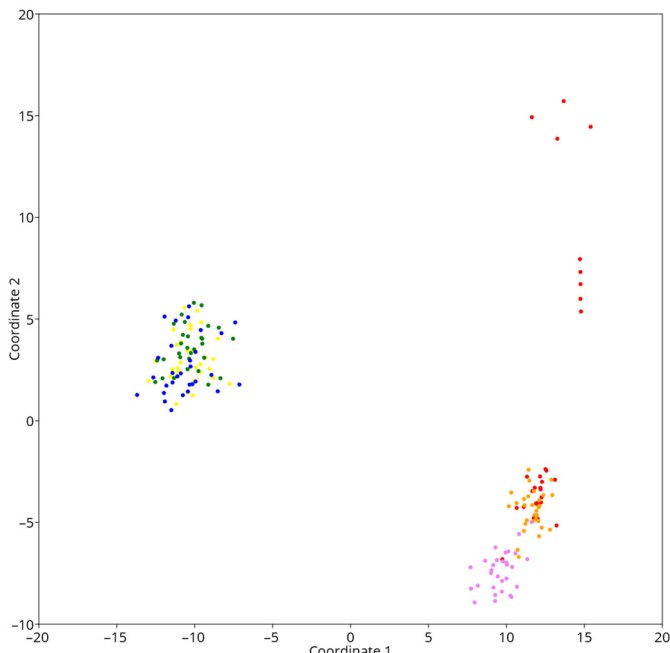

**Figure 5.** Clustering of *L. sibirica* trees under study using the UMAP method. Cluster I: Ls_Tl—Red, Ls_Ih—Orange, Ls_Kh—Violet. Cluster II: Ls_Bn—Yellow, Ls_Kr—Green, Ls_Gn—Blue.

There was a division of populations into two clusters. The first cluster includes mountain larch populations (*Ls_Tl*, *Ls_Ih*, *Ls_Kh*), and the second cluster includes lowland populations (*Ls_Bn*, *Ls_Kr*, *Ls_Gn*).

Populations of *P. obovata*. At the bottom are the genetic distances according to Ney ($D_N$) and at the top are the $F_{ST}$ values.

Populations of *L. sibirica*. At the bottom are the genetic distances according to Ney ($D_N$) and at the top are the $F_{ST}$ values.

## 4. Discussion

### 4.1. Genetic Diversity and Genetic Originality of P. obovata and L. sibirica Populations

Molecular genetic analysis of populations of two species of coniferous plants located in the Northern and Middle Urals showed that genetic diversity indicators are significantly (*p*-value < 0.050) higher for the total sample of *L. sibirica* ($P_{95}$ = 0.886; *He* = 0.218; $n_e$ = 1.426; *I* = 0.329) when compared to *P. obovata* ($P_{95}$ = 0.774; *He* = 0.146; $n_e$ = 1.344; *I* = 0.230).

*P. obovata* in the Northern and Middle Urals has a lower genetic diversity, since it is distributed evenly over a large area, while *L. sibirica* is fragmented in the study region.

To determine genetic diversity at the population level, rare DNA fragments (occurring with a frequency of less than 5%) are important. In the total sample of *L. sibirica*, only 6 (5.3% of the total number of DNA fragments) were noted, and in the total sample of *P. obovata*, 19 rare DNA fragments (16.5%) were noted. According to Yu. A. Yanbaev and Z. Kh. Shigapov [60], the presence of a large percentage of rare DNA fragments identified using isoenzymes in the Southern Urals (from 16.2 to 30.8%) confirms the introgressive hybridization of closely related spruce species. Our data on the number of rare DNA fragments (19 pcs or 16.5%) identified using ISSR markers confirm Yu. A. Yanbaev's and Z. Kh. Shigapov's hypothesis of a large percentage of rare DNA fragments during hybridization of closely related species *P. obovata* and *P. abies* in the region of study.

In order to maintain genetic diversity during reforestation, it is necessary to create sustainable plantations that are not inferior to, nor exceed, the level of genetic diversity of the original, natural populations [38].

In each of the two studied species, populations with the highest and lowest indicators of genetic diversity were identified, but the differences between populations with maximum

and minimum indicators were not significant. In *P. obovata*, the maximum indicators of genetic diversity were noted in the *Po_Ch* population ($He$ = 0.178; $n_a$ = 1.675; $n_e$ = 1.295; $I$ = 0.276), and in *L. sibirica*, in the *Ls_Gn* population ($He$ = 0.243; $n_a$ = 1.693; $n_e$ = 1.414; $I$ = 0.364).

Identification of populations of long-lived woody plant species with maximum levels of genetic diversity is not sufficient to make recommendations for selecting populations for the purpose of conserving genetic diversity and collecting seeds for reforestation. If some parts of the species range differ in the frequency ratio of alleles and haplogroups, then for each specific region it becomes possible to distinguish rare and widely occurring haplogroups, and on the base of the frequency of their occurrence, carry out the procedure of "weighing" the alleles identified in populations using molecular marking [47]. As a consequence, each population within a particular region can be characterized in terms of the proportion of rare and common alleles in the form of a calculated genetic originality coefficient.

We proposed to use populations with both typical and specific gene pools to preserve the gene pool. For reforestation, seeds must be collected in populations with typical gene pools, that is, in populations that contain typical DNA fragments which are common in the region of study. According to the results of our studies, it was found that among the six populations of *P. obovata* studied, two populations had the minimum GOC values (*Po_Gn* with GOC = 0.554 and *Po_Ch* with GOC = 0.676), that is, these two populations are carriers of typical alleles for the study region and have typical gene pools. In addition, the *Po_Ch* population had the highest genetic diversity.

In *L. sibirica*, the lowest GOC values were found in the *Ls_Bn* (GOC = 0.372) and *Ls_Kr* (GOC = 0.432) populations. These two populations have typical gene pools and have typical alleles. And the highest rates of genetic diversity in *L. sibirica* were noted in another population—*Ls_Gn*.

*4.2. Genetic Structure and Differentiation of Six Populations of P. obovata and Six Populations of L. sibirica*

When analyzing the genetic structure of six *P. obovata* populations, it was found that the index of subdivision of population differentiation ($G_{ST}$) for the total sample of *P. obovata* was 0.331.

According to the literature, interpopulation differences in isoenzyme markers in spruce are 9.9% [26]. According to Yu. A. Yanbaev, the intergenetic variability ($F_{ST}$) of spruce in the Southern Urals varied from 3.9 to 4.1 according to the polymorphism of allozyme markers, and the research of Z. Kh. Shigapova [60] shows that 7.1% of the genetic diversity falls on the interpopulation component. Interpopulation differences for microsatellite loci are 9.7%, and 2.9% for mitochondrial loci in the southern and northern groups of European populations [32]. At the same time, a strong genetic differentiation of European and Siberian spruce populations from Eastern Europe to the Russian Far East on the base of the second mtDNA nad1 intron was noted. The interpopulation component of the genetic diversity of the studied specimens is 65%, while the intrapopulation variability is 35%. This is associated with the abrupt border of geographical distribution of haplotypes of Northern European and Siberian spruce passing beyond the Urals.

To obtain more complete and objective information on the genetic diversity and degree of intraspecific differentiation of Siberian spruce populations, it is advisable to use genetic markers of different types of inheritance and levels of variability.

Our data on genetic differentiation within the Northern and Middle Urals are consistent with the data obtained by E. Mudrik et al., but over a much larger area. This phenomenon is explained by the fact that our studies were carried out in the zone of introgression of two species, affecting the Northern and Middle Urals.

Genetic distances ($D_N$) between *P. obovata* populations varied from 0.052 to 0.217. In general, the genetic distances between the studied populations of *P. obovata* were small. However, genetic distances determined using other types of markers have been smaller.

Thus, genetic distances using allozyme markers in Central Siberia did not exceed 0.01 [28]. Comparison of these indicators is incorrect since these are markers of different types of inheritance.

There were no significant correlations between genetic and geographic distances ($R^2 = 0.0529$, $p = 0.16$). The same pattern was established using allozyme markers in Central Siberia [28]. However, there was a significant positive correlation between genetic and climatic distances ($R^2 = 0.5576$, $p = 0.03$). This can be explained by the fact that, due to the complex relief of the Ural Mountains, climatic differences between populations contribute much more to differentiation than the distance between them.

According to the UMAP and dendrogram data, the studied spruce populations were divided into four groups. Three are formed by the populations *Po_Kr*, *Po_Br*, and *Po_Os*, and the fourth combines trees from the populations *Po_Ch*, *Po_Gn*, and *Po_Kg*. This can be explained by the fact that the *Po_Br* and *Po_Os* populations grow in a zone of high anthropogenic pressure and under conditions of habitat fragmentation. And the *Po_Ch*, *Po_Gn*, and *Po_Kg* populations grow in similar conditions. In addition, they are historically connected by logging routes, which could lead to increased gene flow between populations and reduced differentiation between them. In addition, the genetic diversity of populations can be strongly influenced by random genetic drift, which may result in erosion of genetic diversity through the loss of rare alleles [61,62].

An analysis of the genetic structure of six populations of *L. sibirica* showed that, for the total group of *L. sibirica*, the index of subdivision of populations ($G_{ST}$) in the case of using needles was 0.177, which was lower than that of spruce populations. An indicator of a high level of differentiation (*He*) was also observed using SSR markers [41]. Also, interpopulation differentiation in other studies has ranged from 6 to 7%, for *L. gmelinii* in the Far East it has reached 8% [63,64]. Research on *L. decidua* demonstrated a near-total lack of intraspecific differentiation among the examined Romanian samples [63]. Similarly, investigations into the genetic structure of larch in the Urals, employing isozyme markers, unveiled a lesser degree of differentiation (6.1%) when compared to ISSR analysis. This divergence could be attributed to the focus on distinct genomic components during the two analyses [35]. Genetic distances ($D_N$) between *L. sibirica* populations varied from 0.028 to 0.149. In general, the genetic distances between the studied populations of *L. sibirica* were small and lower than those in *P. obovata*. This may be due to the fact that larch samples, compared to spruce samples, were located more compactly and mainly in the Northern Urals.

For populations of Siberian larch, there was an average positive correlation between genetic and geographic distances ($R^2 = 0.3574$, $p < 0.05$). As well as a significant positive correlation of genetic and climatic distances ($R^2 = 0.8685$, $p < 0.05$). Probably, climatic differences affecting the pollination time of trees in separate populations contribute to the differentiation to a greater extent. In addition, climatic variation is determined not only by the latitudinal-longitudinal distribution, but also by the peculiarities of the environment, particularly by the closeness of the Ural Mountains.

According to the UMAP and dendrogram data, the examined populations of larch were divided into two groups. The first cluster included mountain larch populations (*Ls_Tl*, *Ls_Ih*, *Ls_Kh*), and the second cluster included lowland populations (*Ls_Bn*, *Ls_Kr*, *Ls_Gn*). At the same time, within the groups, the samples as a whole were poorly differentiated from each other.

Thus, taking into account the coefficient of genetic originality and differentiation of populations, it is recommended to collect seeds from populations containing DNA fragments typical for the study region, in *P. obovata, Po_Gn* with GOC = 0.554, and in *L. sibirica, Ls_Bn* with GOC = 0.372. These populations were in the same cluster with other populations of the corresponding plant species growing in the study area. At the same time, within the cluster, there was low differentiation between populations of the same plant species, and, taking into account the typical gene pools of the selected populations, the seed material from them is suitable for reforestation in the study region.

In addition to identifying unique accessions from populations of *P. obovata* and *L. sibirica* for seed collection, our studies are important for selecting populations with different contents of various natural bioactive compounds, to improve the efficiency of forest resource use. The established ISSR haplogroups typical for *P. obovata* and *L. sibirica* populations, as well as rare and specific ISSR amplicons, can be used to determine the geographical origin of *P. obovata* and *L. sibirica* wood, which will serve as a basis for detecting and suppressing illegal logging, as well as improve control over the transportation of timber.

## 5. Conclusions

The analysis of the genetic diversity of two coniferous species in the Middle and Northern Urals (*P. obovata* and *L. sibirica*) revealed that Siberian larch populations are characterized by greater genetic diversity than Siberian spruce populations, which may be related to the fragmentation of the Siberian larch range and contrasting growing conditions. When studying populations of *P. obovata*, 19 rare DNA fragments (16.5%) were found, as well as a significant degree of interpopulation differentiation, which may be a consequence of introgressive hybridization of closely related spruce species in the study area. The data on genetic diversity, genetic originality, and subdivision of populations of two conifer species obtained in the course of the study can be used to select reproductive material for reforestation purposes, in the search for natural bioactive compounds, taking into account the differentiation of populations in the study region, as well as for the conservation of relevant forest resources.

**Supplementary Materials:** The following supporting information can be downloaded at: https://www.mdpi.com/article/10.3390/f14091822/s1, Supplementary Table S1. Pairwise geographic distances (km) between the studied samples of *P. obovate*; Supplementary Table S2. Pairwise geographic distances (km) between the studied samples of *L. sibirica*; Supplementary Table S3. Morphological characterization of *P. obovata*, *P. abies* and *L. sibirica*; Supplementary Table S4. Characterization of DNA fragments of 6 populations of *P. obovate*; Supplementary Table S5. Characterization of DNA fragments of 6 populations of *L. sibirica*; Supplementary Table S6. Calculation of the genetic originality coefficient (GOC) on the example of 6 populations of *P. obovata* based on the polymorphism of ISSR markers; Supplementary Table S7. Calculation of the genetic originality coefficient (GOC) on the example of 6 populations of *L. sibirica* based on the polymorphism of ISSR markers; Supplementary Figure S1. The band profiles with ISSR primer X9 (ACC)6G for the samples of *P. obovata* from the populations of Gainy's forestry (*Po_Gn*); Supplementary Figure S2. The band profiles with ISSR primer M1 (AC)$_8$CG for the samples of *P. obovata* from the populations of Krasnovishersk's forestry (*Po_Kr*); Supplementary Figure S3. The band profiles with ISSR primer M1 (AC)$_8$CG for the samples of *P. obovata* from the populations of Gainy's forestry (*Po_Gn*); Supplementary Figure S4. The band profiles with ISSR primer M1 (AC)$_8$CG for the samples of *P. obovata* from the populations of Cherdyn's forestry (*Po_Ch*); Supplementary Figure S5. The band profiles with ISSR primer X9 (ACC)6G for the samples of *P. obovata* from the populations of Cherdyn's forestry (*Po_Ch*); Supplementary Figure S6. The band profiles with ISSR primer X9 (ACC)6G for the samples of *P. obovata* from the populations of Krasnovishersk's forestry (*Po_Kr*); Supplementary Figure S7. The band profiles with ISSR primer X9 (ACC)6G for the samples of *P. obovata* from the populations of Karagai's forestry (*Po_Kg*); Supplementary Figure S8. The band profiles with ISSR primer X9 (ACC)6G for the samples of *P. obovata* from the populations of Berezniki's forestry (*Po_Br*); Supplementary Figure S9. The band profiles with ISSR primer M3 (AC)8CT for the samples of *L. sibirica* from the populations of Gainy's forestry (*Ls_Gn*); Supplementary Figure S10. The band profiles with ISSR primer CR-215 (CA)6GT for the samples of *L. sibirica* from the populations of Cherdyn's forestry (*Ls_Bn*); Supplementary Figure S11. The band profiles with ISSR primer ISSR8 (GAG)6C for the samples of *L. sibirica* from the populations of Krasnovishersk's forestry (*Ls_Kr*); Supplementary Figure S12. The band profiles with ISSR primer X10 (AGC)6C for the samples of *L. sibirica* from the populations of Vishera Nature Reserve (*Ls_Ih*); Supplementary Figure S13. The band profiles with ISSR primer X11 (AGC)6G for the samples of *L. sibirica* from the populations of Vishera Nature Reserve (*Ls_Tl*); Supplementary Figure S14. The band profiles with ISSR primer X11 (AGC)6G for the samples of *L. sibirica* from the populations of Vishera Nature Reserve (*Ls_Ih*); Supplementary Figure S15. The

band profiles with ISSR primer X10 (AGC)6C for the samples of *L. sibirica* from the populations of Karpinsk's forestry (*Ls_Kh*); Supplementary Figure S16. The band profiles with ISSR primer M3 (AC)8CT for the samples of *L. sibirica* from the populations of Krasnovishersk's forestry (*Ls_Kr*); Supplementary Figure S17. The band profiles with ISSR primer CR-215 (CA)6GT for the samples of *L. sibirica* from the populations of Gainy's forestry (*Ls_Gn*); Supplementary Figure S18. Graph of dependence of genetic ($D_N$) and geographical distances of *P. obovata* samples; Supplementary Figure S19. Mantel test for the WorldClim data and genetic ($D_N$) distance of *P. obovata* samples; Supplementary Figure S20. Graph of dependence of genetic ($D_N$) and geographical distances of *L. sibirica* samples; Supplementary Figure S21. Mantel test for the WorldClim data and genetic ($D_N$) distance of *L. sibirica* samples.

**Author Contributions:** Data curation, S.B. and N.C.; formal analysis, and investigation, N.C., L.Z. and Y.N.; methodology, R.K.; resources, S.B.; supervision, S.B. and R.K.; validation, S.B. and A.Z.; statistical analysis, V.P. and Y.N.; writing—original draft, S.B., R.K. and N.C.; writing—review and editing, S.B. and R.K. All authors have read and agreed to the published version of the manuscript.

**Funding:** This study was funded within the framework of state assignment No. FSNF-2023-0004 of the Federal State Autonomous Educational Institution for Higher Education "Perm State National Research University" in science and by the Government of Perm Krai, research project No. C-26/776 dated 31 March 2022. Open access funding was provided by University of Helsinki (Finland), including the Helsinki University Library, via R.K.

**Data Availability Statement:** Data presented in this article are available on request from the corresponding author.

**Acknowledgments:** Open access funding provided by University of Helsinki.

**Conflicts of Interest:** The authors declare no conflict of interest.

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
