# Peer review of "Genetic Uniqueness and Genetic Structure of Populations of Picea obovata Ledeb. and Larix sibirica Ledeb. in the Northern and Middle Urals"

_forests, doi:10.3390/f14091822_

Round 1

Reviewer 1 Report

Dear authors,

I attached a pdf file with my comments in comment boxes and highlighted in text. 

Author Response

Authors’ response: We thank to review for their work done and the great help provided to the authors in the work on the manuscript.

Q1: 2: Authors used only two coniferous species to their study. Therefore, better to use species name in title.

Authors’ response:  article title changed to:

Genetic uniqueness and genetic structure of populations of Picea obovata Ledeb. and Larix sibirica Ledeb. in the Northern and Middle Urals

Q2: 19: Author should mention the which molecular markers that are used to detect genetic diversity

Authors’ response: added ISSR markers:

“In this study, we present an innovative approach for selecting populations suitable for reforestation, taking into account their genetic uniqueness using Inter Simple Sequence Repeats (ISSR) markers.”

Q3: 31: Arrange alphabetical order

Answer: we arranged Keywords: genetic differentiation; genetic originality; genetic structure; Inter Simple Sequence Repeats (ISSR); Larix; Picea; typical alleles; Ural

Q4: 178: Morphological characteristics of three species can be presented in a table form.

Authors’ response:

Morphological characterization of P. obovatа, P. abies [1] и L. sibirica [1,2]

â„–

Trait / tree

Picea

obovata Ledeb. siberian spruce

Picea

abies (L.) Karst. – european spruce

Larix sibirica Ledeb – siberian larch

1.

Tree height, crown type

Up to 30 meters, crown narrow-conic

30 to 50 meters, crown sharp conic

Up to 45 meters, crown conic (young trees) to broad and irregular (old trees)

Bark

Gray, rimose

Reddish-brown to gray

Gray, thick, rimose

Needles

Linear awl-shaped tetraquetrous needles

Linear awl-shaped tetraquetrous needles

Narrow, linear, soft needles

Young cones

Purple-red, single, elongated ovoid, 13-20 mm long, 6-7 mm wide

Elongated cylindric, bright red to green

Pink to red, 8-12 mm long

Mature cones

Brown, elongated ovoid, 5-8 cm long.

Brown, elongated cylindric, 10-16 cm long, 3-4 cm wide

Brown to gray-black, 2-3 cm long

Seeds

Dark brown, obovoid, 4 mm long, 2.5 mm wide, seed scale 10-13 mm long

Dark brown, ovoid, sharp-pointed, 4 mm long, seed scale yellow-red, 12 mm long

Dark brown, ovoid, 2 mm long, seed scale yellow-brown

  1. Ovesnov, S.A.E., E.G.; Kozminykh, T.V.; Baranova, O.G.; Kamelin, R.V.; Kovtonyuk, N.K.; ... Yagontseva, T.A. Illustrated guide to plants of the perm region. Perm, Russia: Book world, 2007.
  2. Dylis, N.V. Larch. Moscow, USSR: Forest industry, 1981.

Q5: 211: Author should mention, how they designed above ISSR primers. If these primers selected from previous study, proper reference should be inserted. Authors should mention, why they used two different sets of primers for Siberian spruce and Siberian larch.

223: What is the reason for using 3 different primers for each species?

Authors’ response: we have rewritten the sentence and added a reference to the article:

The ISSR primers were designed by Kalendar et al. [54], and previously selected effective ISSR primers for Pinus sylvestris and P. sibirica (Table 2) were used [30].

54           Kalendar, R.; Schulman, A.H. Transposon-based tagging: Irap, remap, and ipbs. Methods in Molecular Biology 2014, 1115, 233-255. doi:10.1007/978-1-62703-767-9_12.

Effective ISSR primers have been previously also identified for P. obovata [35], and for L. sibirica [63].

We found two primers common to the two studied species (Picea obovate Ledeb. and Larix sibirica Ledeb.): CR-215 with the formula (CA)6GT), and the other had three nucleotides: X10 with the formula (AGC)6C). These species belong to different genera (Picea and Larix). These species have different genomes, including tandem repeats, which include microsatellites. Therefore, 3 primers effective only for Picea obovate Ledeb. (M1 with formula (AC)8CG; CR-212 (CT)8TG; X9 (ACC)6G, Table 3) and other 3 primers effective only for Larix sibirica Ledeb. (M3 with formula (AC)8CT; X11(AGC) 6G; ISSR8 (GAG) 6C) were determined.

Q7: 268: Move to Materials and method section

Authors’ response: Thank you, we have moved this block of text to the methods chapter.

Reviewer 2 Report

I think the paper by Andrei Zhulanov et al. has sound methods and conclusions and I have only two minor comments. However, I'm concerned whether the paper has big enough scope for the journal. If no one else raises this question than I agree with publishing the paper after minor revisions. But if other reviewers and/or editor feel that the paper, which covers genetic diversity and differentiation of a sample of 6 populations per species from two tree species, with the samples covering only a subset of natural range, is too small-scale then I second that position and would recommend publishing the paper in a more local journal.

76-82: A reference for the range would be preferred.

83-84: Is it most common? The references 14 and 15 don't seem to mention this, if I didn't overlook it, and if they are even meant for this sentence. Pinus may be even more widespread. Reference or rewording is needed.

129: The reference number 30 seems to be misplaced.

Author Response

Q1: 76-82: A reference for the range would be preferred.

Authors’ response: We've added references to Picea abies – P. obovata spruce complex:

It is pointed out that it is difficult to draw the exact boundary of their distribution, since many morphological features smooth out towards the periphery of the range; the term Picea abies – P. obovata spruce complex is currently used [16-19].

  1. Podchong, S.; Schmidt-Vogt, D.; Honda, K. An improved approach for identifying suitable habitat of sambar deer (cervus unicolor kerr) using ecological niche analysis and environmental categorization: Case study at phu-khieo wildlife sanctuary, thailand. Ecological Modelling 2009, 220, 2103-2114.
  2. Orlova, L.; Gussarova, G.; Glazkova, E.; Egorov, A.; Potokin, A.; Ivanov, S. Systematics and distribution of spruce species in the north-west of russia. Dendrobiology 2020, 84, 12-29.
  3. Tsuda, Y.; Chen, J.; Stocks, M.; Kallman, T.; Sonstebo, J.H.; Parducci, L.; Semerikov, V.; Sperisen, C.; Politov, D.; Ronkainen, T., et al. The extent and meaning of hybridization and introgression between siberian spruce (picea obovata) and norway spruce (picea abies): Cryptic refugia as stepping stones to the west? Mol Ecol 2016, 25, 2773-2789.
  4. Hall, D.; Olsson, J.; Zhao, W.; Kroon, J.; Wennstrom, U.; Wang, X.R. Divergent patterns between phenotypic and genetic variation in scots pine. Plant Commun 2021, 2, 100139.

Q2: 83-84: Is it most common? The references 14 and 15 don't seem to mention this, if I didn't overlook it, and if they are even meant for this sentence. Pinus may be even more widespread. Reference or rewording is needed.

Authors’ response: "most common" has been corrected to “common”:

“Species of the genus Larix Mill. are common woody plants in Russia and the planet as a whole.”

We moved the references to these works elsewhere and added the following:

  1. Li, W.; Manzanedo, R.D.; Jiang, Y.; Ma, W.; Du, E.; Zhao, S.; Rademacher, T.; Dong, M.; Xu, H.; Kang, X., et al. Reassessment of growth-climate relations indicates the potential for decline across eurasian boreal larch forests. Nat Commun 2023, 14, 3358.

Q3: 129: The reference number 30 seems to be misplaced.

Authors’ response: We moved the reference elsewhere and used a different one:

The highest level of species diversity in the Urals was noted in the “Permian-Kama Cis-Uralian” population, which was established using a complex of data from morphological and isoenzyme analyzes [47].

47           Putenikhin, V.P.F., G.G.; Shigapov, Z.K. Methods for maintaining genetic heterogeneity when creating artificial "populations" of forest-forming species. Conifers of the boreal zone 2007, 24, 272-278.
